# Global coral genomic vulnerability explains recent reef losses

Oliver Selmoni [1,2,3,4,5] ✉, Phillip A. Cleves[2,3,4,9] ✉ &
Moises Exposito-Alonso [1,5,6,7,8,9] ✉

The dramatic decline of reef-building corals calls for a better understanding of coral adaptation to ocean warming. Here, we characterize genetic diversity of the widespread genus *Acropora* by building a genomic database of 595 coral samples from different oceanic regions—from the Great Barrier Reef to the Persian Gulf. Through genome-environment associations, we find that different *Acropora* species show parallel evolutionary signals of heat-adaptation in the same genomic regions, pointing to genes associated with molecular heat shock responses and symbiosis. We then project the present and the predicted future distribution of heat-adapted genotypes across reefs worldwide. Reefs projected with low frequency of heat-adapted genotypes display higher rates of *Acropora* decline, indicating a potential genomic vulnerability to heat exposure. Our projections also suggest a transition where heat-adapted genotypes will spread at least until 2040. However, this transition will likely involve mass mortality of entire non-adapted populations and a consequent erosion of *Acropora* genetic diversity. This genetic diversity loss could hinder the capacity of *Acropora* to adapt to the more extreme heatwaves projected beyond 2040. Genomic vulnerability and genetic diversity loss estimates can be used to reassess which coral reefs are at risk and their conservation.

Climate change-induced heat waves and subsequent coral bleaching have reduced reef-building corals worldwide by ~14% just over the 2009–2018 decade[1]. As corals form the physical scaffold of the reef ecosystem, their decline also threatens the persistence of dependent wildlife—estimated to be one quarter of all known marine species[2]. Despite this mass mortality, several lines of research indicated that coral populations are evolutionarily adapted to local environments (*e.g.*, cross-breeding experiments[3,4], colony genotyping from thermally contrasted reefs[5–10], and experiments exposing corals to controlled thermal conditions[8,11]), suggesting that evolutionary adaptation is a possible escape route from heat vulnerability. However, global efforts in reef conservation

monitoring and risk assessment are still lacking integrated indicators of coral evolution.

Genotype-environment associations (GEA) are a promising approach for identifying genetic variants underlying coral adaptation to warming oceans[12]. GEA assumes that reefs that have survived past heatwaves are enriched in heat-tolerant coral genotypes. A common GEA approach leverages decades of satellite observations of sea surface temperature to identify reefs with contrasting thermal histories[13]. Corals are sampled from such reefs and genotyped to identify adaptive signals—genetic variants predominantly found in heatwave-exposed populations[14]. The relationship between heatwave trajectories and adaptive signals can then be synthesized in environmental genomics

[1]Department of Plant Biology, Carnegie Institution for Science, Stanford, CA, USA. [2]Department of Embryology, Carnegie Institution for Science, Baltimore, MD, USA. [3]Department of Biology, Johns Hopkins University, Baltimore, MD, USA. [4]Department of Molecular and Cell Biology, University of California Berkeley, Berkeley, CA, USA. [5]Department of Integrative Biology, University of California Berkeley, Berkeley, CA, USA. [6]Department of Biology, Stanford University, Stanford, CA, USA. [7]Department of Global Ecology, Carnegie Institution for Science, Stanford, CA, USA. [8]Howard Hughes Medical Institute, University of California Berkeley, Berkeley, CA, USA. [9]These authors jointly supervised this work: Phillip A. Cleves, Moises Exposito-Alonso.
✉e-mail: oliver.selmoni@gmail.com; pacleves@berkeley.edu; moisesexpositoalonso@gmail.com

models, and these models can be used to generate predictive maps of adaptive genotypes distribution[7]. When combined with future climate projections, these maps help pinpoint reefs at risk of genomic vulnerability[15]—where predicted heat-adaptive genotypes distribution may not match future thermal exposure (also known as genomic offset or maladaptation).

However, GEA approaches are notoriously prone to false discoveries[16], making predictive models based on unvalidated adaptive signals potentially unreliable. These false discoveries arise from neutral genetic forces such as connectivity and genetic drift, which can create spatial allele frequency patterns that coincidentally correlate with thermal histories[14]. A proposed solution to this problem is to compare GEA results across different species; if the same adaptive signal appears in multiple species, it is less likely to be a random correlation[17]. This approach has recently been applied to plants, successfully identifying robust candidate core genes involved in climate adaptation[18]. While these shared signals likely represent only a subset of corals' full adaptive potential, their robustness could enhance the reliability of spatial predictions of heat-adaptive genotype distribution, providing critical insights for conservation planning[12].

Here, we focus on the widespread *Acropora* coral genus and build a multi-species genomic dataset to identify evolutionary signals of parallel local adaptation to past heatwave exposure. We then use these signals to explain recent coral loss and build a global map of coral genomic vulnerability.

## Results and discussion

### Global patterns of *Acropora* sp. genomic diversity

To generate a broad picture of coral genomic diversity, we compiled genomic sequencing datasets from over one thousand corals, of which 595 samples passed our quality controls. These samples include five species of the genus *Acropora* sp.: *A. downingi* from the Persian Gulf[19], *A. digitifera* from the Ryukyu Archipelago (Japan)[20], *A. cervicornis* from the Florida Reef Tract[21], *A. tenuis* from the Australian Great Barrier Reef (GBR)[10], and *A. millepora* from the GBR[22] and New Caledonia[5] (Fig. 1A, Table S1). For every dataset, researchers sampled coral colonies from multiple sampling sites (5–22 sites) across a large spatial scale (> 100 km), and genotyped each colony with whole genome or restriction–site-associated DNA sequencing. For comparisons between species, we assembled a consolidated genotype matrix by mapping reads from each dataset against the genome of *A. millepora*[9] (NCBI RefSeq GCF_013753865.1; v. 2.1), which was the only chromosome-level assembly available for the genus at the time of analysis (2022-2024). To map reads from different *Acropora* species to the *A. millepora* reference, we used a read mapping method allowing for sequence divergence[23] that achieved high mapping rates in all datasets (mean percentage of mapped reads in each dataset were between 89% and 95%; Fig. S2B). Next, we called biallelic single-nucleotide polymorphisms (SNPs) after accounting for uncertainty due to low sequencing coverage[24]. The resulting multi-species genotype matrix was analyzed using three methods summarizing genetic diversity patterns across datasets. First, a Principal Coordinate Analysis (PCoA) on SNPs shared between datasets revealed genetic patterns coherent with taxonomy and geography between the *Acropora* sp. worldwide after correcting for batch effects (Fig. 1A, Fig. S7, Table S2). Specifically, the first PCoA axis separated Atlantic corals from Indo-Pacific corals [*i.e.*, Florida Reef Tract *A. cervicornis* separated from the other samples, percentage of variance explained (PVE) = 44%]. The second PCoA axis separated Coral Sea *A. millepora* (from New Caledonia and GBR) from GBR *A. tenuis*, Japan *A. digitifera*, and Persian Gulf *A. downingi* (PVE = 29%). Second, the mean nucleotide genetic diversity of *Acropora* sp. was π/bp = 0.37% (dataset averages ranging 0.14–0.91%; Table S1A), which was consistent with previous *A. millepora* estimates[9]. Third, genetic diversity within datasets followed a scaling relationship with area[25] that confirmed

decreasing gene flow with increasing geographic distance, as observed in other *Acropora* populations (Fig. S17)[26,27].

### Overlapping genomic signals of heat-adaptation across *Acropora* sp

Given that controlled experiments exposing different *Acropora* individuals to heat show heritable differences in heat tolerance[3,4,8,11], we reason that surviving colonies from reefs that suffered heatwaves in the past would be enriched in genotypes enabling their survival now. To test this hypothesis with the compiled datasets, we conducted GEA along the genome using a latent factor mixed model (LFMM2)[28]. First, we characterized past heatwaves as a proxy for the environmental selection using sixteen variables derived from satellite time series. These sixteen variables described short-term (10 years before sampling) and long-term (since 1985) trends (averages, standard deviations, maximal values) in Sea Surface Temperature, Degree Heating Week (DHW), and Sea Surface Temperature Anomaly (SSTA) (Fig. S8)[29]. Next, using GEA analysis, we characterized the association between each of these environmental variables and SNPs genome-wide, correcting for confounding population structure. We finally used Picmin[17] to identify 10-kb genomic windows significantly enriched ($Q < 0.1$) in heat-associated SNPs in three or more datasets. Such overlapping signals were detected in all datasets, with their number proportional to the SNP count per dataset (Table S1C). Of the sixteen environmental variables, long-term maximal DHW (measured since 1985, $DHW_{max}$; Fig. 1A) was the one resulting in the largest number of overlapping signals——85 in total, with 55 found in 3 datasets, 27 in 4 datasets, and 3 in 5 datasets (Fig. 1B, Fig. 1C, Table S1C, Supplementary Data 1). DHW is a variable describing the accumulation of thermal stress above the local SST baseline, with values above 8 °C-week associated with severe, widespread bleaching[30]. Genotype-environment association studies conducted on a single species are prone to false discoveries[31]. While it is not possible to entirely eliminate these false positives, the overlap across species makes the shared adaptive signals more robust[17,18]. These shared signals likely represent important candidate genotypes with adaptive potential and may reflect molecular convergence underlying heat adaptation across species[17].

### Signals of heat-adaptation point to heat shock responses genes

To identify molecular functions potentially impacted by these heat-associated SNPs, we analyzed gene annotations within the 85 genomic windows with overlapping signals of adaptation to $DHW_{max}$ (Fig. 1D, Supplementary Data 1, Table S3). We ran a gene ontology (GO) enrichment analysis of the 119 annotated genes[32], and retrieved the genes' transcriptomic profiles during heat stress from a meta-analysis of different *Acropora* sp. experiments[33]. The second and third most enriched GO terms were "heat shock protein binding" and "unfolded protein binding" (SetRank FDR-adjusted $P = 0.002$; Table S3). Five genes are annotated with molecular functions associated with heat shock responses and protein folding: Heat Shock 70kDa protein (HSP70), Protein SSUH2 homolog, Uromodulin (all up-regulated in heat-exposed *Acropora* sp.), Protein unc-45 homolog B (down-regulated), and Fibronectin (non-differentially expressed). HSP70 is a molecular chaperone that refolds proteins misfolded by heat stress[34], and higher levels of a heat shock protein 70 were found to be associated with bleaching resistance in field experiments[35]. As heat-associated bleaching causes corals to lose the symbiotic algae living inside their tissue that are essential for their survival, it seems plausible that this symbiosis would be a target of selection during heatwave exposure[36]. Consistent with this hypothesis, we found two significantly enriched GO terms referring to the Inositol signalling pathway ("inositol monophosphate 1-phosphatase activity" and "phosphatidylinositol-4-phosphate phosphatase activity", SetRank FDR-adjusted $P = 0.006$), which have been proposed to be involved in the maintenance of a functional coral-algal symbiosis[37,38], and that underlie key

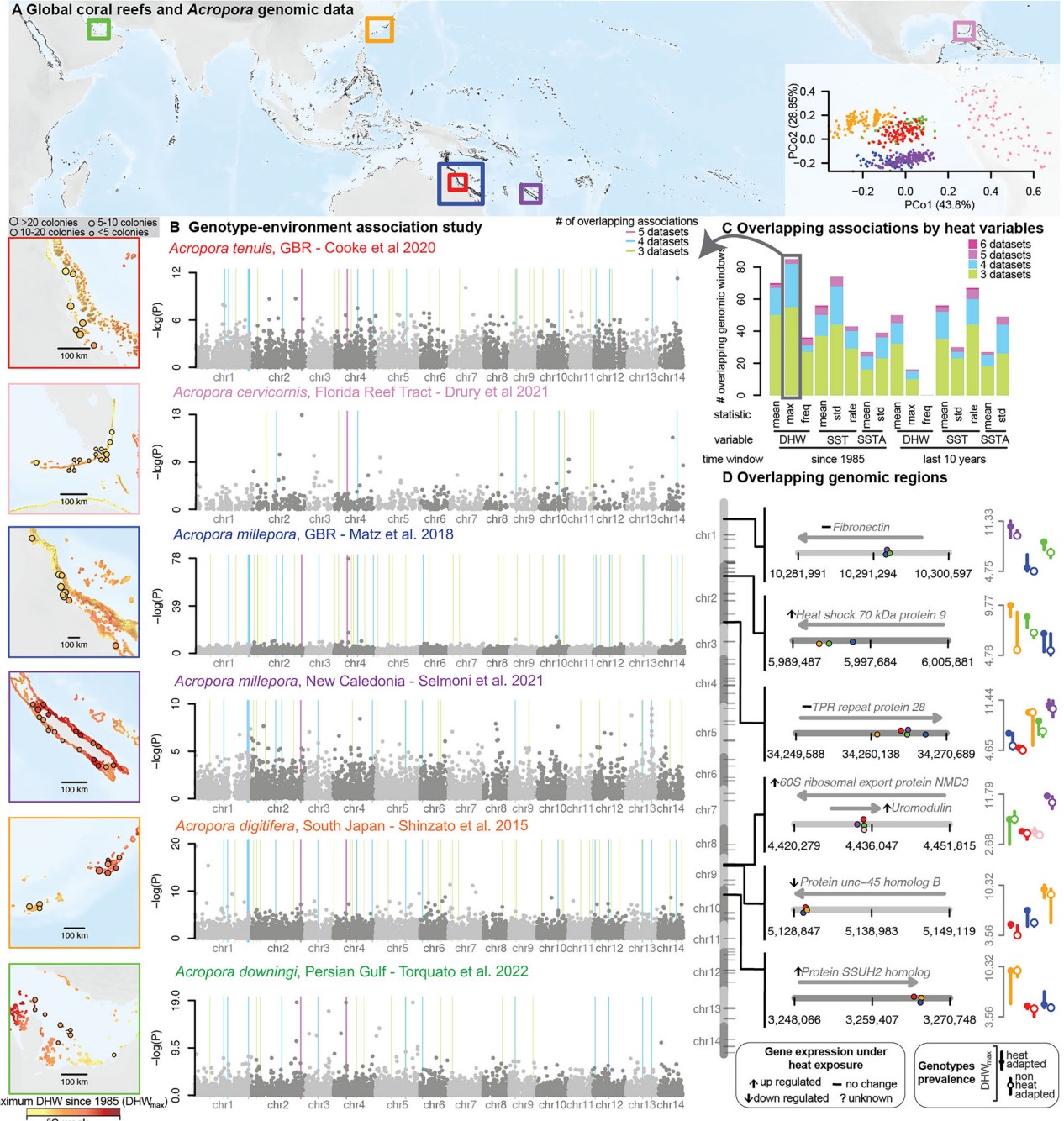

**Fig. 1 | Genome-wide scans of evolutionary signals in 595 *Acropora* sp. identifies overlapping signals of local heat adaptation. A** Distribution of global coral reefs and DNA sequencing data from six datasets describing different *Acropora* sp. populations (squares)[5,10,19–22]. For every dataset, shown are the sampling locations across reefs exposed to contrasting thermal history [*i.e.*, maximum Degree Heating Week ($DHW_{max}$) since 1985 (maps on the left)]. The inset shows the two main principal coordinate axes summarizing the genetic distances between the six *Acropora* populations ($N = 595$). **B** Genomic overlap analysis of $DHW_{max}$-associated genotypes across datasets. The six Manhattan plots show the significance [-log($P$)] of the association between $DHW_{max}$ and SNP variation genome-wide (x-axis) for each dataset. Background lines indicate genomic windows significantly enriched ($Q < 0.1$) in $DHW_{max}$-associated SNPs across three (green), four (blue), or five (purple) datasets. **C** Results of the overlap analysis for sixteen heat stress descriptors. These descriptors are defined by (1) the time window covered (1985 to the sampling year or the ten years prior), (2) the heat stress variable measured over time (DHW = maximum annual Degree Heating Week, SST = maximum annual sea

surface temperature, SSTA = mean annual sea surface temperature anomaly), and (3) the statistic used to summarize heat stress trends (mean, standard deviation [std], frequency of years with DHW > 4 °C-week [freq], rate of yearly change [rate]). For each heat stress descriptor, the bar plot shows the count of genomic windows significantly enriched ($Q < 0.1$) in heat-stress-associated SNPs, categorized by the number of datasets enriched at the same windows. **D** Six of the overlapping genomic regions, showing the genomic position of genetic variants associated with $DHW_{max}$ (circles, colored by *Acropora* population) and annotated genes (arrows). Whether the annotated genes are known to change expression during heat stress from a meta-analysis of RNAseq of various *Acropora* species is shown next to the gene name[33]. For every significant genetic variant, the prevalence of heat-adapted and non-heat-adapted genotypes by $DHW_{max}$ is shown for *Acropora* populations (colors). Background maps were produced using the Global Relief Model from NOAA (ETOPO 2022[82]) and the Global Distribution Map of Coral Reefs from the UNEP World Conservation Monitoring Center[83].

metabolomic changes in heat-exposed *Acropora aspera*[39,40] and *Pocillopora aliciae*[41]. In addition, we identified three significant GO terms (SetRank FDR-adjusted $P = 0.004$) referring to glutamate receptor activity, which is another relevant term for symbiosis since glutamate metabolism is central to coral-algal nutrient cycling and is destabilised under heat-exposure[42]. Results from GO enrichment analyses should be interpreted cautiously and not viewed as conclusive. Nevertheless, they can be valuable for highlighting prominent candidate genes for further validation. Given their association with molecular heat responses and symbiosis, the genes underlying these functional enrichments should be prioritized in molecular and genetic studies to determine their roles in heat tolerance, symbiosis, and bleaching[12].

### Predicting the spatial distribution of heat-adapted *Acropora*

We then wanted to evaluate how well temporal patterns of heatwaves exposure could predict the frequency of *Acropora* candidate adaptive genotypes across reef regions. For each sampling site, DHW time series were decomposed by 5-year windows, and $DHW_{max}$ was calculated for each window. We then built a Random Forest model[43] using time-decomposed $DHW_{max}$ to predict the frequency of candidate heat-adaptive genotypes, retrieved from the 85 genomic windows with overlapping adaptive signals (Fig. 2A). This environmental genomic model can predict the frequency of heat-adapted genotypes in new unsampled regions with a mean absolute error (MAE) of 13% (SD = ± 6%, in a leave-one-dataset-out cross validation; Fig. 2B). Since the environmental genomics model is built on adaptive signals shared across *Acropora* species, it likely overlooked adaptive genotypes that have evolved within single species. We therefore questioned whether predictive power would be higher in an alternative model built on all adaptive signals, including those found in single datasets only. We found that this alternative model had lower predictive power (MAE = 21 ± 16%) compared to the environmental genomic model built on overlapping adaptive signals (Fig. S12). The lower predictive power could be due to the higher uncertainty in adaptive signals that are not shared across species. As coral sampling and sequencing efforts expand[44], this environmental genomic model could be updated with data from other *Acropora* species to improve its predictive power. Analogous models could also be developed for coral taxa with different life history traits and thermal tolerance (e.g., *Pocillopora*, *Porites*)[45,46], as these taxa might be selected differently by heatwave exposure.

### Spatial predictions of heat adaptation explain local *Acropora* decline

Although there is considerable interest in monitoring the conservation status of reefs, the observed variation in recent coral cover loss is notoriously challenging to interpret with heatwave data alone[47,48]. We thus asked whether predictions from our environmental genomic model could help explain local *Acropora* decline[47,48]. For this proof-of-concept analysis, we leveraged in situ data from 166 sites across the Indo-Pacific, where *Acropora* cover had been repeatedly surveyed between 2012–2018 (Fig. 3A)[49]. We first checked how well spatial autocorrelation and heat stress data (without genetic information) explained changes in *Acropora* cover (Fig. 3B; Table S4A). We found that a mixed model accounting for heatwave exposure ($DHW_{max}$) between surveys explained a small portion of the spatial variation in cover change (marginal $R^2 = 0.02$)[50]. Adding information about past heatwave exposure significantly improved the model fit, but still explained limited spatial variation (marginal $R^2 = 0.07$; $W = 0$, $P < 2.2 \times 10^{-16}$ under a Mann-Whitney-Wilcoxon test comparing jackknife resampled marginal $R^2$). In contrast, spatial variation in *Acropora* cover change was explained substantially better when considering heatwave exposure between surveys and the expected frequencies of heat-adaptive genotypes (predicted from the environmental genomic model; marginal $R^2 = 0.25$; Mann-Whitney-Wilcoxon $W = 0$, $P < 2.2 \times$

$10^{-16}$; Fig. 3B). The increase in the model fit is likely due to the separation of *Acropora* decline rates between reefs by expected frequency of heat-adaptive genotypes (Fig. 3C; Table S4C). For example, reefs with an adaptive-genotype frequency of 0% are expected to undergo an average *Acropora* decline of 2.7% per °C-week of heat exposure. As the predicted frequency of heat-adaptive genotypes increases, this decline is expected to weaken and become non-significant at a genotype frequency of ~70%. Notably, we observed a decline in model fit (marginal $R^2 = 0.11$) when the model accounted for the expected frequency of all candidate adaptive genotypes, including those not shared across species (Fig. S13). These results suggest that better predictions of the vulnerability of reefs to future heatwaves could be generated by including robust genetic information of past local adaptation from a few coral colonies interpolated at larger scales.

### *Acropora* heat adaptation and genomic vulnerability under future climate

Global predictive models of coral decline and real-time detection of coral bleaching are primarily guided by remote sensing of sea surface temperature[13], and do not yet account for the past local evolutionary adaptation and genetic differences between corals from different reefs. As *Acropora* is a geographically widespread reef-building coral taxa (Fig. S14)[51], we used our environmental genomic model to project the spatial distribution of candidate heat-adaptive genotypes at worldwide scale under both present and forecasted heat stress (Fig. 4). The projections spanned 2010-2040 and were based on global $DHW_{max}$-time series, with forecasts predicted using a combination of five climate change models under a moderate emission scenario (SSP2-4.5; similar results under a high emission scenario; Fig. S16)[52]. We then computed a year-by-year categorization of reefs worldwide by heat exposure and probability of heat adaptation. We classified reefs as severely heat-exposed when their annual $DHW_{max}$ exceeded 8 °C-week[30], and as non-heat-adapted when their expected frequency of heat-adaptive genotypes was below 70%---as suggested by the *Acropora* cover change analysis (Fig. 3C). Non-heat-adapted reefs that were severely heat-exposed were deemed to be under genomic vulnerability.

The global proportion of reefs experiencing severe heat exposure is expected to rise sharply under future climate, whereas genomic vulnerability is predicted to increase until around 2030, after which it will stabilize (Fig. 4C). During 2020-2024, for example, 37% of the reefs worldwide were projected to be severely heat-exposed, with 67% of these reefs (25% globally) classified under genomic vulnerability (Fig. 4A). The regions projected with highest proportions of genomic vulnerable reefs were the Caribbean (75%) and the Red Sea (67%). By 2036-2040, the global fraction of reefs under severe heat exposure is expected to reach 72%, with about 51% of these reefs (37% globally) under genomic vulnerability (Fig. 4B). Importantly, a considerable fraction of global reefs (27%) was forecasted with thermal conditions beyond the predictable range of our environmental genomic model, making regional projections in the Red Sea, the Persian Gulf, the Caribbean unattainable. Among the predictable regions, the Western Indian Ocean showed the largest proportion of reefs under genomic vulnerability (92%). These projections suggest that although more coral reefs will be exposed to heatwaves in the coming decade, *Acropora* adaptive potential from pre-existing presumably adaptive genotypes could keep pace with forecasted heat stress in most of the Indo-Pacific—at least until 2040.

While encouraging, these results should not suggest that coral reefs will remain unaffected in the coming years. In heat-exposed *Acropora*, selection for thermally tolerant colonies will likely involve widespread mortality and recolonisation from a few remnant populations, as observed recently in Caribbean *Acropora palmata*[53]. Such turnover is likely to disrupt reef ecosystem functions, especially if other reef-building taxa respond similarly to *Acropora*. Moreover,

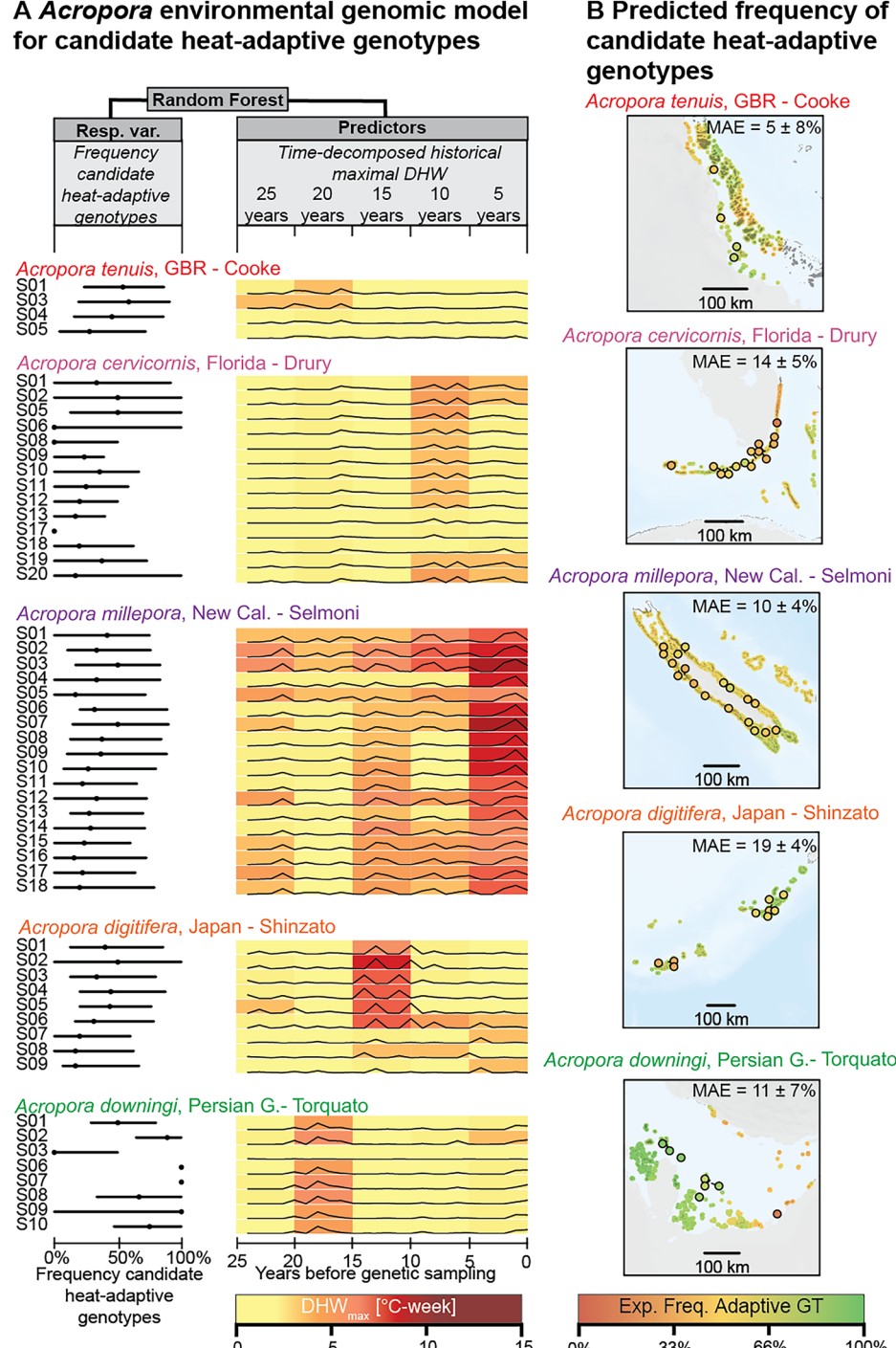

**Fig. 2 | *Acropora* environmental genomic model for candidate heat-adaptive genotypes. A** Environmental genomic model predicting the expected frequency of candidate heat-adaptive genotypes from patterns of past heatwave exposure, described by maximal Degree Heating Week ($DHW_{max}$) measured across 5-year windows. The environmental genomic model was based on five datasets (total number of reefs $N = 53$). For each dataset, the figure shows the frequency of candidate heat-adaptive genotypes for each reef (left side) and the reef thermal history (right side), represented by $DHW_{max}$ values aggregated into 5-year windows spanning from 25 years before sampling to the year of sampling. **B** Spatial predictions of the frequency of candidate heat-adaptive genotypes across the reefs of five *Acropora* datasets. For every dataset, predictions were made using the environmental genomic model trained on data from the other four datasets (i.e., leave-one-out cross validation). The mean absolute error (MAE) of the predicted genotype frequencies is shown in the top right corner of each map. Circles represent real genotype frequencies at sampling sites. Background maps were produced using the Global Relief Model from NOAA (ETOPO 2022[82]) and the Global Distribution Map of Coral Reefs from the UNEP World Conservation Monitoring Center[83].

further research is needed to determine whether local adaptation can happen under heat stress levels that exceed those observed globally today. In this regard, the only reliable solution to maintain healthy coral reefs in the long term is to limit the increase in heat stress by reducing carbon emissions[54].

## Loss of genetic diversity in *Acropora* under global change

Future adaptation to the unprecedented heat forecasted after 2040 will likely be fueled by standing genetic diversity. However, the high mortality induced by some extreme heatwaves might cause the loss of entire *Acropora* populations and the erosion of associated genetic

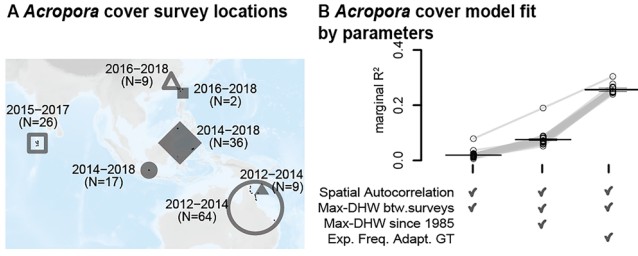

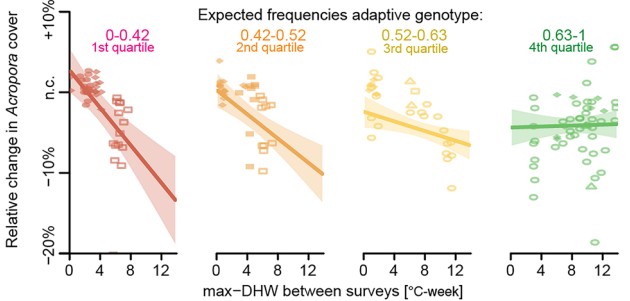

**Fig. 3 | Adaptive genotype predictions explain spatial patterns of *Acropora* decline by heat exposure. A** Map displaying the number of survey sites with repeated measurements of *Acropora* cover across marine provinces of the Indo-Pacific (symbols)[49]. Shown is also the period covered by surveys in every marine province. **B** The distributions of marginal coefficients of determination (marginal R²) for models explaining the change in *Acropora* cover using different explanatory variables: marine province of the survey site (Spatial Autocorrelation), maximal heat stress measured between surveys (Max-DHW btw. surveys), maximal historical heat stress (Max-DHW since 1985), and expected frequency of heat-adaptive genotypes––derived from the environmental genomic model. For every model, the marginal R² distribution (*N* = 163) is shown in a boxplot defined by a central line corresponding to the median value, a box delimited by the lower and upper quartiles, and box whiskers extending to ±1.5 times the interquartile range. Gray lines represent marginal R² estimations from distinct jackknife subsamples. **C** The model with the highest marginal R² from (**B**): the relative change in *Acropora* cover is represented as a function of maximal heat stress between surveys for sites with different expected frequencies of heat-adaptive genotypes (red: 0–0.42%, orange: 0.42–0.52%, yellow: 0.52–0.63; green: 0.63–1). Plot symbols correspond to the marine provinces of the survey site [as displayed in (**A**)]. The shaded areas show the 95% confidence intervals of the regression lines. Background map was produced using the Global Relief Model from NOAA (ETOPO 2022[82]) and the Global Distribution Map of Coral Reefs from the UNEP World Conservation Monitoring Center[83].

diversity. Consequently, *Acropora* might enter a feedback loop that prevents further adaptation and increases population extinction risk[55]. To estimate the amount of *Acropora* genetic diversity at risk by reef area loss, we used the mutations-area relationship (MAR) framework[25], which characterizes the scaling relationship between the number of genetic variants (SNPs) and geographical range across the six *Acropora* datasets. We first applied the MAR framework to each of the six *Acropora* datasets (Fig. S17), and then used a linear mixed model to summarize scaling mutation-area relationships across species (Fig. S18). Validated through stochastic simulations under different extinction scenarios (Fig. S21), this scaling relationship was summarized in an *Acropora* $z_{MAR}$ parameter of 0.31, that can be used to predict the percentage of *Acropora* genetic diversity at risk by reef loss (95% CI = [0.20, 0.42]) (Figs. 4D; S20). For example, during 2020–2024, the global proportion of severely heat-exposed reefs was 37%, a reef area estimated to harbor 13% of the global *Acropora* genetic diversity. Accounting for genomic vulnerability, the proportion of at-risk reefs for the same period is reduced to 25%, corresponding to an instantaneous loss of 8% of the global *Acropora* genetic diversity. These results suggest that quantifying the spread of heat-adapted corals under

future heat will be key for accurately estimating genetic loss and anticipating long-term coral decline.

## Conservation strategies for heat-adapted corals
Our temporal projections suggest that approximately three-quarters of the reefs classified as putatively heat-adapted in 2024 originate from heatwave selection events experienced after 2015 (Fig. 4C), meaning that such reefs may still be recovering reef cover after the recent loss of unadapted corals. However, this recovery might be hampered by local stressors (*e.g.*, destructive fishing or coastal development[56,57]). We therefore compared the 2024 projected distribution of heat-adapted reefs with worldwide proxies of human pressure on reefs—such as the Human Pressure Index (HPI)[56] and maps of marine protected areas[58]. At global scale, heat-adapted reefs were less exposed to local human stressors than non-heat-adapted ones ($\Delta$HPI = −0.076, 95% CI = [−0.080, −0.071], Analysis of Variance $F$(df = 45036,1) = 876, $P < 2.2 \times 10^{-16}$; Fig. S22A), and their proportion was significantly higher inside MPAs (26%), compared to outside (17%; Chi-square test $X^2$(df = 1, N = 45577) = 507, $P < 2.2 \times 10^{-16}$; Fig. 4E). Some regions differed from this global trend: heat-adapted reefs were less protected in the Persian Gulf, and as protected as non-heat-adapted reefs in Australia, East Asia, South Asia (Fig. 4E). Heat-adapted reefs may be the most likely to survive future climate change and thus should be a priority for protection from human impacts by global reef conservation efforts. Recent research proposed that coral conservation efforts should focus on a portfolio of climate refugia (*i.e.*, reef regions escaping heatwaves and cyclones, and with a strong potential for larval dispersal)[59]. As these refugia are by definition regions not impacted by climate change, heat-adapted coral reefs are drastically underrepresented in the climate refugia portfolio (7% heat-adapted reefs inside refugia compared to 27% outside of refugia; Fig. S22C).

These results solidify the notion that evolutionary adaptation is a key component to understanding global change-driven risks and reef losses. Because adaptation depends on standing genetic diversity, it is essential not only to protect heat-adapted genotypes but also to conserve a broad genetic diversity portfolio to support future adaptive potential. We thus advocate that reef conservation priorities should include indicators of coral genomic vulnerability and genetic diversity loss.

## Methods
A visual summary of methods and data used is shown in Fig. S1.

### Datasets selection
We identified genomic datasets from NCBI meeting the following criteria: (1) genotyped at least 50 individual *Acropora* colonies; (2) genotyped colonies collected from at least five distinct sampling locations; (3) sampled locations precisely geo-located and spread across a spatial range of at least 100 km; (4) collected during a temporal window of a maximum of two years; (5) genotyped via high throughput sequencing techniques (whole genome sequencing or restriction sites associated DNA sequencing). We identified seven datasets meeting these criteria: PRJNA593014 (*Acropora millepora* from central Australian Great Barrier Reef - GBR)[9], PRJNA702071 (*Acropora millepora* from New Caledonia)[5], PRJNA665778 (*Acropora cervicornis* from the Florida Reef Tract)[21,60], PRJDB4188 (*Acropora digitifera* from the Ryukyu Archipelago, Japan)[20], PRJEB37470 (*Acropora tenuis* from the central GBR)[10], PRJNA434194 (*Acropora millepora* from the GBR)[22], PRJNA750412 (*Acropora downingi* from the Persian Gulf)[19]. Detailed information on the datasets is reported in Table S1A.

### Sequencing reads quality filtering, mapping and SNPs calling
Raw sequencing reads from each of these seven datasets were downloaded using the fastq-dump tool (SRA toolkit, v. 3.0.0) from the SRA database[61]. We analyzed the quality of raw sequencing reads from every dataset using FastQC (v. 0.11.9)[62] and performed read trimming

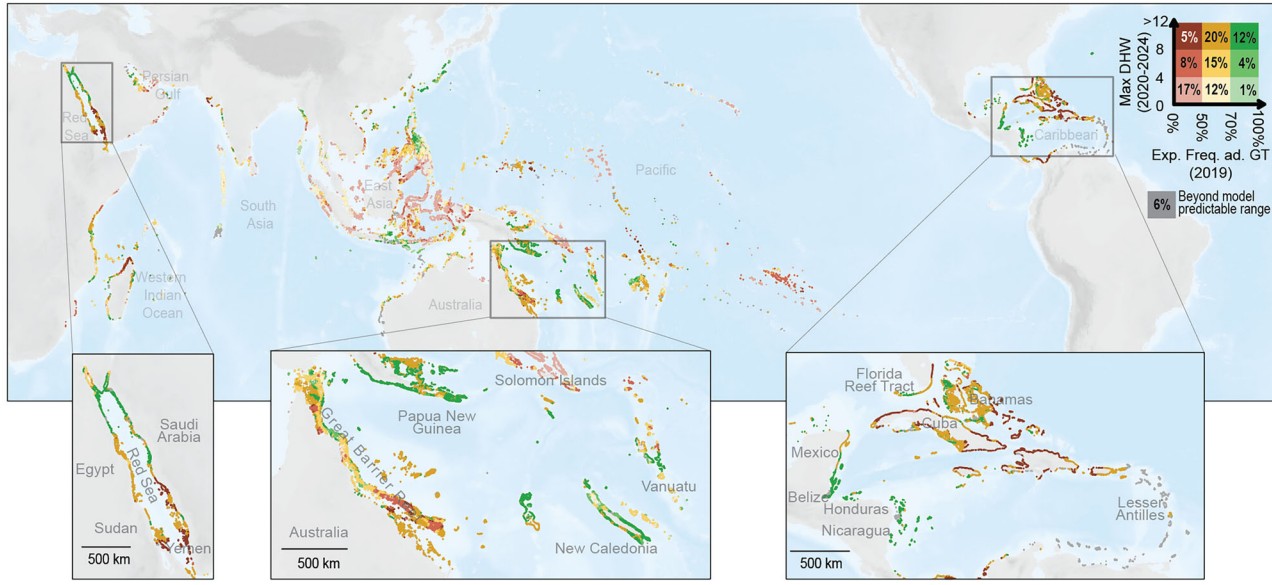

**A** Genomic vulnerability of *Acropora* worldwide: 2020-2024

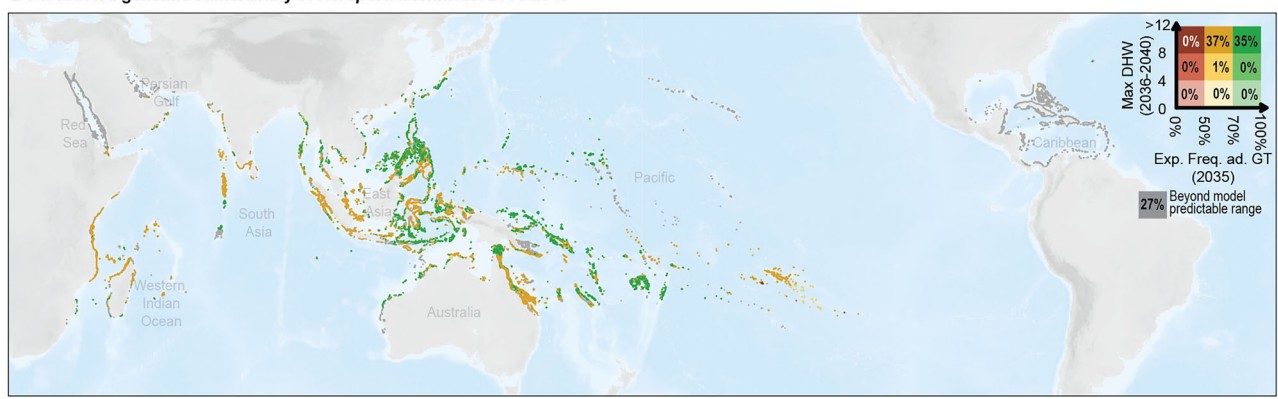

**B** Forecasted genomic vulnerability of *Acropora* worldwide: 2036-2040

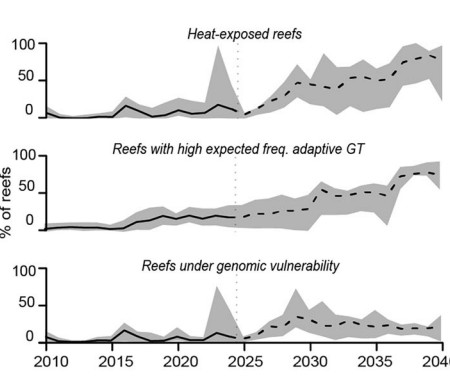

**C** Temporal projections of heat exposure and genomic vulnerability

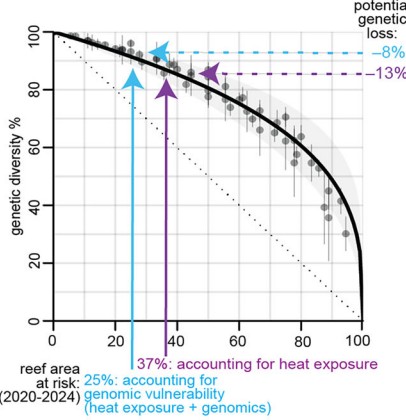

**D** *Acropora* mutations area relationship

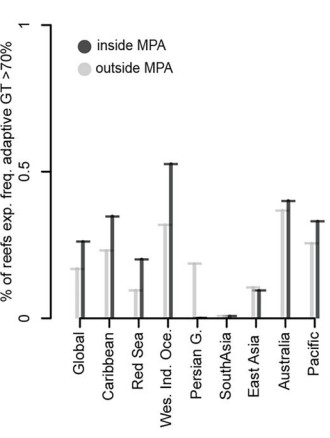

**E** Expected frequency of heat-adaptive genotypes inside protected areas

and adapters clipping using Trim Galore (v. 0.6.1)[63]. Trimmed sequencing reads from every dataset were then mapped to the chromosome-level assembly of *Acropora millepora* (NCBI Refseq: GCF_013753865.1; v. 2.1)[9]. Read mapping was performed using Next-GenMap (v. 0.5.2)[23], a software designed for the alignment of reads even under substantial divergence between the query and the reference genome (*e.g.*, the reference genome of a different species). We ran NextGenMap using the default settings, and then evaluated the mapping statistics using Samtools (v. 1.15.1; Fig. S2)[64].

Aligned reads from every dataset were processed using tools from the ANGSD suite (v. 0.938)[24]. First, we estimated the genotype likelihood of polymorphic loci (*P* threshold of $10^{-6}$) using the GATK model[65], then we computed the posterior genotype probabilities and called hard genotypes for biallelic single-nucleotide polymorphisms (SNPs, probability calling threshold: 95%). For every dataset, we investigated the distribution of variant sites along the genome by counting the number of SNPs per 50 kbs genomic windows. The low-coverage WGS Acropora millepora dataset from central GBR[9] showed a

**Fig. 4 | Present and future projections of genomic vulnerability across global reefs.** Global map of coral reefs colored by expected frequency of heat-adaptive genotypes based on heat stress before 2020 (0–50%: red, 50–70%: yellow, 70–100% green). The intensity of colors represents the intensity of heat stress [maximal Degree Heating Week (DHW$_{max}$)] during 2020–2024 (low: <4 °C-week, significant: ≥4 and <8 °C-week, severe: ≥8 °C-week). In grey, reefs are exposed to heat stress levels beyond the predictable range of the environmental genomics model. **B** Global map of expected frequency of heat-adaptive genotypes based on heat stress before 2035. The intensity of colors represents the intensity of fore-casted heat stress during 2036-2040. **C** Year-by-year projected proportions of reefs exposed to severe heatwaves (top graph), of reefs with an expected frequency of heat-adaptive genotypes above 70% (middle graph) and of reefs under genomic vulnerability (*i.e.*, reefs exposed to severe heatwaves and having an expected fre-quency of heat-adaptive genotypes below 70%; bottom graph). Full lines represent projections from past heat stress data, and dotted lines from forecasted heat stress

under a moderate climate change scenario (SSP2-4.5)[52]. Shaded areas in the back-ground display the range of projections across ten oceanic regions. **D** Estimated mutations-area relationship (MAR) of the *Acropora* populations (black line, shaded area as interval of confidence). Points represent the mean genetic diversity loss per area loss (bars represent the 95% confidence interval), as observed in stochastic simulations of extinction across all datasets (*N* = 58). Solid arrows indicate the proportion of global reef area at risk of extinction according to heatwave exposure (blue) and genomic vulnerability (purple)---based on 2020-2024 projections. For these potential losses of reef area, dashed lines indicate the corresponding genetic loss. **E** Mean proportion of reefs (*N* = 45,577) with expected frequency of heat-adaptive genotype above 70% inside (black) and outside (grey) marine protected areas (MPA), globally and by oceanic region (genotype frequencies predicted in 2024). Background map was produced using the Global Relief Model from NOAA (ETOPO 2022[82]) and the Global Distribution Map of Coral Reefs from the UNEP World Conservation Monitoring Center[83].

highly heterogeneous distribution of SNPs across the genome and was excluded from downstream analyses (Fig. S3). The number of variant sites yielded by the four RADseq datasets was substantially lower (9,303 to 64,947 SNPs per dataset), compared to WGS datasets (A. tenuis: 1.3 M SNPs, A. digitifera: 23 M SNPs). To facilitate downstream comparisons between datasets, we reduced the WGS datasets to 100,000 SNPs using a custom Python script that applied a stratified subsampling of SNPs across the genome. Detailed statistics of sequencing read filtering and SNP calling are reported in Fig. S2 and Table S1B.

### Genotype matrices filtering
We removed outlier samples from every dataset using a three-step procedure. The first step focused on identifying outliers based on sequencing and alignment statistics (i.e., number of reads, percentage of mapped reads, genome coverage, sequencing depth, base quality score, mapping score, number of duplicate reads, and GC content; Figs. S2, S4). The second step of the filtering procedure was based on a PCA of the genotype matrices (Fig. S5), aiming to identify groups of outliers that might correspond to groups of genetically isolated indi-viduals (*e.g.*, cryptic species, isolated reefs). In the third step of the filtering procedure we filtered out highly correlated samples (R > 0.9; Fig. S9), which might correspond to clones. Table S1B summarizes the number of outliers samples filtered out at every step.

Pairwise nucleotide diversity (π) was calculated for every sam-pling site (minimum number of samples per site >5) using ANGSD, focusing on 10 kbs genomic windows having reads in at least 3 data-sets. We then used a generalized random factor model (R-package glmmTMB, v. 1.1.2)[66] to describe nucleotide diversity while accounting for variation across datasets, sampling sites and genomic windows. Rarefied allelic richness was calculated on the filtered genotype matrices using the R-package hierfstat (v. 0.5)[67].

We evaluated the degree of genetic relatedness among samples from all datasets using a two-step approach. First, we retrieved the number of SNPs shared between any pair of datasets (median number of shared SNPs: 45 [IQR = 30], Fig. S6). Second, we computed the genetic distance between samples from all datasets---i.e., the per-centage of non-identical genotypes across the shared SNPs (Fig. S6). The genetic distance across all samples was then summarized in a principal coordinate analysis (PCoA) (Fig. 1A). Finally, we assessed how geography, taxonomy and dataset explained the variation across the genetic PCoA axes (Table S2) using linear mixed models in the glmmTMB R package (v. 1.1.2)[66].

### Environmental data
We characterized heat stress gradients by processing three environ-mental variables from the Coral Reef Watch (CRW)[30]: maximal annual Sea Surface Temperature, maximal annual Degree Heating Week (DHW) and mean annual Sea Surface Temperature Anomaly (SSTA).

These variables are commonly used to explain geographical patterns of coral bleaching and decline[47,48,68]. DHW represents the accumulation of thermal stress (*i.e.*, temperature >1 °C above the climatological maximal mean temperature) over a 12-weeks period. DHW above 4 °C-week is associated with significant coral bleaching, and DHW above 8 °C-week with severe, widespread bleaching[69]. SSTA is the difference between SST measured at a given reef on a specific day of the year, and the reef's climatological SST for that day of the year. For every sam-pling site in the genomic datasets, we extracted the year-by-year values of the three CRW variables from 1985 until the year of sampling. We then summarized the temporal trends of these variables using a total of eight statistics:

For SST, we calculated the overall temporal average and standard deviation, as well as the average rate of SST change per year.

For DHW, we calculated the overall temporal average and maximum value, and the frequency of years with a DHW > 4 °C-week.

For SSTA, we calculated the overall temporal average and standard deviation.

In addition to these long-term environmental descriptors, we created a set of short-term descriptors by applying the same calcula-tions on a temporal window of ten years prior to sampling. The var-iation of the eight long-term and eight short-term variables across datasets is shown in Fig. S8.

### Genotype-environment association analyses
Genotype-environment association (GEA) analyses were run using a two-step framework. This framework was repeated independently for each of the 16 variables characterizing historical patterns of heat stress, as described below. In the first step we ran GEA analysis within each dataset. We first applied standard filters to the genotype matrices to remove SNPs with rare allelic variants (minor allele frequency <5%), SNPs with one genotype having very high frequency (major genotype frequency > 95%), and we also discarded SNPs and samples with high missing rates (>20%; Fig. S9). The GEA analysis was then run using the Latent Factors Mixed Models approach (LFMM)[28]. The optimal number of latent factors was estimated for every dataset using the *SNMF* function of the LEA R package (v. 2.8.0; 5 repetitions, optimal number of latent factors reported in Table S1C, Fig. S10)[70]. We then used the *impute* function of the LEA package to impute missing SNPs in the genotype matrix of every dataset. Finally, we associated SNPs variation across individuals with heat stress variables at sampling sites using the *lfmm* function of the lfmm R package (v. 1.1)[28]. In the second step of the framework, the results of the GEA analyses were compared between datasets using PicMin[17]. PicMin compared the genome-wide distribu-tion of LFMM-adjusted *P* (*i.e.*, calibrated to control for genomic infla-tion factor; Table S1C) across datasets. Specifically, 10-kb genomic windows were ranked in each dataset based on their LFMM-adjusted *P* (with top ranks corresponding to windows containing SNPs with low,

i.e., significant, $P$). PicMin then tested for enrichment of top-ranked windows across datasets and assigned an enrichment $P$ to each genomic window, which was then corrected for false discoveries (q-value method[71]). Genomic windows with significant overlap were those showing (1) an enrichment $Q < 0.1$ and (2) overlap of significant GEA associations in at least three datasets.

## Validation of GEA on an independent dataset

We retrieved the genotype matrix characterizing central GBR *A. millepora*, sampled from 12 reefs in 2017[9]. This genotype matrix includes 6,386,121 SNPs resulting from the genome-wide imputation of 190 low-coverage samples. We retrieved maximal DHW characterizing past heat stress at every sampling site using the methods described in the Environmental Data section. Next, we computed the association between SNPs within the 85 overlapping genomic windows and maximal DHW using a standard linear regression. Since this *A. millepora* population did not show particular genetic structure[9], correction in the GEA analysis was not necessary. We focused on the 85 genomic windows with overlapping adaptive signals for $DHW_{max}$, and identified the SNP with the lowest GEA $P$ in every window. As a comparison, we identified the lowest GEA $P$ in 1000 randomly selected genomic windows of the same size. We finally used a Wilcoxon rank sum test to compare the distribution of GEA $P$ in the overlapping windows *versus* the random windows (Fig. S11).

## Candidate genes annotation

We retrieved the predicted protein sequences of every gene annotated in the *A. millepora* reference genome and conducted a similarity search (blastp; v. 2.7.1)[72] against the Uniprot database[73] (as accessed on December 1st, 2022). The targets of the similarity search were manually reviewed and annotated metazoan proteins (*i.e.*, labeled with status "Swiss-Prot"). We then focused on the 85 genomic windows with overlapping adaptive signals for long-term maximal DHW, and identified genes located within each window (Supplementary Data 1). Heat stress expression data for these genes were retrieved from an *Acropora* sp. transcriptomics meta-analysis[33] (differential expression threshold: $P = 0.01$). Enrichment analysis of gene annotations was performed using the R package SetRank (v. 1.1)[32], and focused on the Gene Ontology terms describing Molecular Functions. We ran the enrichment analysis with a $P$ threshold of 0.05 and a FDR-adjusted $P$ threshold of 0.05 (Table S3).

## Environmental genomic model

The environmental genomic model focused on candidate heat-adaptive genotypes from the 85 genomic windows repeatedly associated with maximal DHW. For each of these genomic windows, we first retrieved the SNP the most strongly associated with maximal DHW (*i.e.*, lowest LFMM $P$), and identified the SNP's genotype that was prevalent at the highest levels of maximal DHW (that is, the candidate heat-adaptive genotype). Repeated for every dataset, this procedure identified a total of 288 candidate heat-adaptive genotypes. For each of these genotypes, we calculated the frequency of occurrence by sampling site. Next, we split measurements of maximal historical DHW by five-year windows (0–5 years, 5–10 years, 10–15 years, 15–20 years, and 20–25 years before sampling; Fig. 2A). We then build the following random forest model:

$$\text{AGT}_{rgd} = rf(d, g, \text{DHW}_r^{0-5}, \text{DHW}_r^{5-10}, \text{DHW}_r^{10-15}, \text{DHW}_r^{15-20}, \text{DHW}_r^{20-25})$$

where:

$AGT_{rgd}$ is the frequency of an adaptive genotype from a specific genomic window ($g$), a specific sampling site ($r$), and a specific dataset ($d$),

$rf$ is the Random Forest function built from the ensemble of decision trees using as predictors the dataset ($d$), the genomic window ($g$),

and time-decomposed $DHW_{max}$ at the sampling site for 5-year temporal windows ($\text{DHW}_r^{0-5}$, $\text{DHW}_r^{5-10}$, $\text{DHW}_r^{10-15}$, $\text{DHW}_r^{15-20}$, $\text{DHW}_r^{20-25}$).

The model was built on genetic and environmental data from all the *Acropora* datasets, except the *A. millepora* dataset from the GBR[22]––which was excluded because sampled in 2002 and therefore not allowing us to reconstruct the previous 25 years of heat exposure. We evaluated the predictive power of the model using a leave-one-dataset-out cross-validation: we iteratively trained the model using data from all datasets except one, and estimated the predicted frequency of heat-adaptive genotypes in the dataset excluded from the training set. The predictive power was calculated as the mean absolute error (MAE) between real and predicted frequencies of adaptive genotypes at every sampling site (Fig. 2B).

Finally, we investigated whether these models could achieve similar predictive power when built using candidate adaptive genotypes identified solely through single genotype-environment association studies, without overlap analyses across datasets. To do this, we extracted SNPs associated with significant genotype-environment relationships from each dataset, applying three different LFMM FDR thresholds (0.01, 0.1, and 0.2). Models built on these SNP sets consistently exhibited significantly higher MAE (ANOVA $P < 0.01$) compared to the random forest model based on overlapping adaptive signals (Fig. S12).

## Linking environmental genomic predictions with Acropora cover change

*Acropora* cover data across the Indo-Pacific was accessed from the standardized reef surveys of the Catlin Seaview project[49]. We retrieved 166 transects with repeated *Acropora* cover surveys during the 2012-2018 period, spread across 66 distinct reefs from 7 marine provinces [Central Indian Ocean Islands (26 transects), Northeast Australian Shelf (64 transects), South China Sea (2 transects), South Kuroshio (9 transects), Sunda Shelf (17 transects), Tropical Southwestern Pacific (9 transects), Western Coral Triangle (36 transects); Fig. 3A]. For every transect, we then calculated the relative change in *Acropora* cover between the first and the last survey (performed 2 to 5 years apart, depending on the region). Using the same methods described in the "Environmental data" section, we retrieved year-by-year maximal DHW values since 1985 for every transect site. For every transect site, we then calculated:

(1) the maximal DHW between the years of the first and the last survey,

(2) the historical maximal DHW between 1985 and the year of the first survey, and

(3) the 5-years windows time-decomposed maximal DHW for the 25 years before the first survey, then converted into the expected frequency of heat-adaptive genotypes using the environmental genomic model.

The variation of *Acropora* cover over time was assessed using a set of linear mixed models built using the R package glmmTMB (v. 1.1.2)[66]. We first constructed null models where the relative change in *Acropora* cover was explained by variables not related to heat stress, such as the number of years between surveys, or by random factors characterizing spatial autocorrelation at different geographical scales: reef, marine ecoregion, marine province, and marine realm[74]. We then built three alternative models, including different combinations of heat-related variables (Fig. 3B):

Model1) $ACC_{srp} = \beta_0 + \beta_1 DHW_s + u_r + v_p$

Model2) $ACC_{srp} = \beta_0 + \beta_1 DHW_s + \beta_2 DHW_s^{past} + \beta_3 DHW_s DHW_s^{past} + u_r + v_p$

Model3) $ACC_{srp} = \beta_0 + \beta_1 DHW_s + \beta_2 AGT_s + \beta_3 DHW_s AGT_s + u_r + v_p$

where:

$ACC_{srp}$: is the relative change in *Acropora* cover between at given survey site (*s*), located on a specific reef (*r*), and on a specific marine province (*p*);

$\beta_O$ is the fixed intercept;

$\beta_1 DHW_s$ is the effect of recent heat stress---i.e., $DHW_{max}$ measured between surveys at the sampling site (*s*);

$\beta_2 DHW_s^{past}$ is the effect of historical heat stress---i.e., $DHW_{max}$ measured between 1985 and the first survey at sampling site *s*;

$\beta_3 DHW_s DHW_s^{past}$ is the interaction effect between recent and historical heat stress at sampling site *s*;

$\beta_2 AGT_s$ is the effect of the expected frequency of heat-adaptive genotypes at sampling site *s*;

$\beta_3 DHW_s \ AGT_s$ is the interaction effect between recent heat stress and the expected frequency of heat adaptive genotypes at sampling site *s*;

$u_r$ is the intercept random effect of reef *r*;

$v_p$ is the intercept random effect of marine province *p*.

In addition, we built an alternative version of Model 3, where AGT was estimated from candidate adaptive genotypes identified as significant SNPs (LFMM FDR < 0.2) in any dataset (without overlap analysis).

The quality of fit of the different models was assessed by using a jackknife resampling approach (MuMIn R package, v. 1.43)[75] to estimate the distributions of the Akaike Information Criterion (AIC), and of the coefficients of determination [marginal ($mR^2$);i.e., variation explained by fixed factors, and conditional ($cR^2$);i.e., variation explained by fixed and random factors][50]. AIC and $R^2$ distributions were compared across models using the Mann-Whitney-Wilcoxon test[76], and are reported in Fig. S13. The three alternative models showed conditional $R^2$ values (~0.3) similar to those of the null models, suggesting that incorporating thermal histories and genotype frequencies does not provide additional explanatory power for the spatial autocorrelation between reefs and marine provinces. However, the models differed in their marginal $R^2$ values, indicating that certain variables are better at explaining spatial variability in *Acropora* cover change.

## Global projections of genomic vulnerability

We retrieved the geographical distribution of coral reefs reported to a 5 km$^2$ cells grid[77] using the Reef Environment Centralized Information System[29]. We excluded reef cells where *Acropora* corals are not present [according to the Ocean Biodiversity Information System (OBIS), accessed via the R package robis (v. 2.11.3)[78]; Fig. S14]. Past heat stress conditions across the reef cells were characterized using the same methods described in the "Environmental Data" section, using Coral Reef Watch maximal annual DHW from 1985 to 2024. Future heat stress conditions (2025-2100) were accessed from the Coral Reef Bleach Risk Prediction Portal (https://coralbleachrisk.net)[52], which provides worldwide year-by-year forecasts to 2100 of annual maximal DHW. We extracted reef cells' future DHW predictions from a multimodel-averaged forecasts (called "ensemble-5") under two Shared Socioeconomic Pathways (SSP) scenarios: SSP2-4.5 and SSP5-8.5. The environmental genomic model was then used to predict year-by-year expected frequency of heat-adaptive genotypes across reef cells worldwide. We chose 2010 as the starting year of our projections because the environmental genomic model requires 25 years of past heat stress records (CRW data is available since 1985). We chose 2040 as the end year for our projections because, by that time, most of the world's reefs are expected to experience heat stress levels beyond the range used to train the environmental genomic model (Fig. S15). For every year between 2010 and 2040, we calculated:

(1) the fraction of reef cells exposed to severe heatwaves (annual maximal DHW > 8 °C-week);

(2) the fraction of reef cells with a high expected frequency of heat-adaptive genotypes (≥70%, hereafter referred to as heat-adapted reef cells);

(3) the fraction of reef cells under genomic vulnerability (annual maximal DHW > 8 °C-week and expected frequency of heat-adaptive genotypes <70%), and

(4) the year-by-year fraction of reef cells that were available for predictions (*i.e.*, reef cells not exceeding the model's DHW training range).

All these fractions were calculated under both SSP scenarios at the global and regional scale (Fig. S16).

## Mutations area relationship

Mutations-Area Relationship (MAR) were calculated using the methods described in Exposito-Alonso (2020)[25], applied separately to the six *Acropora* datasets. In brief, the raw genotype matrix was preliminarily filtered for missing observations (filtering threshold: 20%, by individual and by SNP; Table S1D)---but not for allele frequencies. Focusing on 100 randomly selected SNPs from the genotype matrix, we repeated 50 times a subsampling procedure that consisted in:

(1) drawing a square of random size across the spatial range of the dataset,

(2) retrieving the sampling sites falling within this square region, and

(3) calculating the number of mutations across the colonies from such sampling sites.

The subsampling procedure was then replicated with ten different subsets of SNPs, yielding a total of 500 subsamples per dataset. We then summarized the mutation-area scaling across datasets using the following a Bayesian linear mixed model [R-package MCMCglmm (v. 2.30)[79]]:

$$log(M)_{dgs} = \beta_0 + zMARlog(A)_s + tSD_d + ud + vg + w_s$$

where:

$log(M)$ is the mutations count in the spatial subsample (*s*) of a given dataset (*d*), under a given genomic subsample (*g*);

$\beta_O$ is the fixed intercept;

$z_{MAR}log(A)$ is the effect of the log-transformed area of the spatial subsample (*s*) on the mutations count;

$tSD_d$ is the effect of the mean sequencing depth of the dataset (*d*) on the mutations count;

$u_d$ is the intercept random effect of the dataset (*d*);

$v_g$ is the intercept random effect of genomic subsample (*g*);

$w_s$ is the intercept random effect of the geographic position of the spatial subsample (*s*).

These explanatory variables were included in the model after a stepwise construction to optimize the goodness of fit (according to the deviance information criterion - DIC; Table S5). This final model returned the $z_{MAR}$ parameter - i.e., the coefficient of regression between log-transformed area and number of mutations - which summarizes the strength of the MAR. The $z_{MAR}$ estimated for *Acropora* was 0.14 [IC = 0.07–0.19] (Fig. S17). Two datasets showed a $z_{MAR}$ significantly lower or higher than the overall *Acropora* estimate: *A. tenuis* from the GBR (z-mar=0.06 [IC = 0.00–0.11]) and *A. millepora* from New Caledonia (z.mar=0.23 [IC = 0.16-0.28]), respectively. As a comparison, we also calculated the $z_{MAR}$ expectation in each dataset under a panmictic population scenario. This was done by calculating MAR using a simulated genotype matrix, where individual genotypes were sampled from an allele frequency spectrum expected under panmixia (Fig. S19). The simulated panmictic $z_{MAR}$ were systematically lower than the real z-mar (Fig. S18).

We then corrected the estimated *Acropora* $z_{MAR}$ for the potential bias due to an uneven number of sampling sites between datasets. This correction was performed using a rarefaction analysis in the nlme R-package (v. 3.1)[80] (Fig. S20). We used the corrected $z_{MAR}$ to estimate theoretical rates of *Acropora* genetic loss by reef habitat loss. To validate these theoretical predictions, we computed stochastic simulations of extinction, where sampling locations were progressively removed from every dataset following distinct spatial patterns (Fig. S21). The validated *Acropora* $z_{MAR}$ was then used to estimate the potential genetic loss at reefs at risk of local extinction, according to the 2020-2024 projections of genomic vulnerability (Fig. 4A).

### Linking genomic vulnerability with global conservation

Reef-cells from our global projections were compared with a set of global descriptors of reef conservation status:

The local human pressure on coral reefs (HPI), previously estimated by Andrello and colleagues[56].

The global distribution of Marine Protected Areas from the World Database of Protected Areas[58]

The global portfolio of proposed coral reefs climate refugia[59].

Focusing on reefs with high probability of heat-adaptation according to our 2020 projections, we evaluated the variation of HPI (analysis of variance) and compared the proportion of reefs inside and outside Marine Protected Areas and climate refugia (Chi-square test; Fig. S22).

### Data availability

Genomic data used in this study are publicly available in NCBI, for the full list of accession numbers and data links please see Supplementary Table 1. Processed data are available at Zenodo[81] (https://doi.org/10.5281/zenodo.10838947). Supplementary Data 1 displays the list of the 85 genomic windows where genotype-environment associations were repeatedly found in different datasets.

### Code availability

Code to reproduce the analysis is available at Zenodo[81] (https://doi.org/10.5281/zenodo.10838947).

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

## Acknowledgements

We are grateful to the openness of many researchers who make genomic data publicly available, making this research possible: Cooke et al., Drury et al., Fulle et al., Matz et al., Selmoni et al., Shinzato et al., and Torquato et al. We also thank the Catlin Seaview Survey project for collecting and giving access to the field survey data for the *Acropora* GBR case study, the Coral Reef Watch for giving access to the degree heating week data, the United Nations Environment Programme World Conservation Monitoring Centre for giving access to the worldwide distribution of coral reefs data, and Dixon et al. for sharing the *Acropora* gene expression data. We thank Rachael Bay and Stephane Joost for early discussions of coral datasets, and thank the MoiLab and Cleves lab for comments and discussions. M.E.-A. is supported by the Office of the Director of the National Institutes of Health's Early Investigator Award (1DP5OD029506-01), the Carnegie Institution for Science, the Howard Hughes Medical Institute, and the University of California, Berkeley. Computational analyses were done on the High-Performance Computing clusters *Memex, Calc,* and *MoiNode* supported by the Carnegie Institution for Science. P.A.C. is supported by an NSF-EDGE grant (2128073), Pew Biomedical and Marine Fellowship (00036631), Revive and Restore, the Carnegie Institution for Science, and Moore Foundation grant (12187). We also thank a Carnegie Venture Grant (P.A.C. and M.E.-A.) for support.

## Author contributions

O.S., P.A.C., and M.E.-A. conceived and led the project. O.S. conducted research, O.S., P.A.C., and M.E.-A. interpreted the results and wrote the manuscript.

## Competing interests

The authors declare no competing interests.
