## [Transparent Peer Review file · Nature Communications]

Global coral genomic vulnerability explains recent reef losses

Corresponding Author: Dr Oliver Selmoni

Version 0:

Reviewer comments:

Reviewer #1

(Remarks to the Author)

In their manuscript, Selmoni, Cleves, and Exposito-Alonso provide an excellent analysis exploring the global genomic vulnerability of reefs using publicly available genomic, photo survey, and environmental datasets. In this study, they perform GEA analyses to understand the interaction between genomic variants and the environment in six datasets covering five species of *Acropora* corals. To identify regions with conserved importance in thermal tolerance across species, they pinpoint genomic windows enriched in significant markers in three or more of the datasets. In corals, few studies have aggregated results across species in this manner to identify conserved thermotolerance markers. Using this conserved set (which includes HSP70, a gene identified in other coral stress studies), the authors build predictive models to estimate the percentage of adaptive genotypes using remotely sensed heat stress data. They then use the expected frequencies of heat-adaptive genotypes, in combination with heat stress data, to predict changes in *Acropora* percent cover, based on photo surveys from the Catlin Seaview Survey. This model is extended to project the genomic vulnerability of reef ecosystems worldwide from 2020–2040 under SSP2-4.5 climate scenarios. Reassuringly, the authors conclude that the proportion of resilient genotypes in populations will increase, although some regions are projected to experience conditions beyond the model's predictive scope (e.g., Caribbean, Persian Gulf, Red Sea). Finally, they overlay regions of predicted resilience and vulnerability onto a global map of marine protected areas (MPAs), finding that heat-resilient reefs were more frequently located within MPAs than outside of them. However, they also found that a large proportion of MPAs are protecting non-resilient reefs, a finding which highlights the need to reassess MPA design to better conserve reefs harboring heat-resilient alleles.

This work is unique and makes a significant contribution to coral research. While lacking experimental data, the study is novel in aggregating results across species to predict reef-level declines. Further experimental work (which is ongoing across the world's tropical reefs) will help ground-truth this modeling framework and improve future projections. The work stands well on its own and its claims are well-supported by the literature and by the rigorous analytical framework outlined in the manuscript. As such, it provides a novel foundation for understanding how at-risk reefs may change in the future.

Flaws in the manuscript are minor and primarily relate to adding language that highlights both the importance and limitations of the work. For example, the "Loss of genetic diversity in *Acropora* under global change" and "Conservation strategies for heat-adapted corals" sections could benefit from more cautionary language. While it is encouraging that allele frequencies will shift toward heat-adaptive variants, I caution that some readers may interpret this to mean that coral reefs will be fine under different climate change scenarios. Shifts in heat tolerance are likely to occur via: A) massive bottleneck events (the more likely scenario, given current emission trajectories), or B) massive reductions in emissions that allow allele frequencies to shift gradually without widespread mortality. Here would be a great place to use Florida *Acropora palmata* as case study – NOAA has an excellent report "Genotypic inventory and impact of the 2023 marine heatwave on *Acropora palmata* (elkhorn coral) populations in the Upper Florida Keys, USA: 2020-2023" by Williams et al. Pretty much all corals have died, and those that happened to survive at one restricted portion of the reef only did so because reef temperatures did not get as high.

I really like the lines "Future adaptation to the unprecedented heat forecasted after 2040 will likely be fueled by standing genetic diversity. However, the high mortality induced by some extreme heatwaves might cause the loss of entire *Acropora* populations and the erosion of associated genetic diversity." I'd like to see this point emphasized even stronger, particularly in the abstract and the concluding sentences of the manuscript. It's extremely encouraging to see that heat tolerance variation exists now, but 2036 is pretty much tomorrow for these corals. Most of these species can technically reproduce at

age 2 but realistically won't until age 5-10 or more. And then from there, if reefs have declined enough, colonies may not be close enough to a viable reproductive partner for their gamete production to even be fruitful. So even if we assume an extremely liberal generation time of 2 years, this is only 5 generations. The remaining genetic diversity will be heat tolerant, but much of the reef matrix that was *Acropora* will be dead.

I have no major methodological concerns with the analysis and believe the manuscript has strong merit for publication as-is. I also appreciate the authors making their code and datasets publicly available. The code is well annotated in the README. However, I have a few minor suggestions for clarification in the methodology:

- 1) I would make sure to note the kind of RAD method used for each of the datasets (rather than just the general term "RAD-Seq"), preferably in the main text, but alternatively in Table S1A. Each RAD method has its own assumptions and the research community has strong opinions on each.
- 2) In the results, it's mentioned that *A. millepora* was used because of the quality of the reference. What defines quality? Contiguity? BUSCO? NCBI annotation? And were there other rationales as well? Was it also because you had two separate *A. millepora* sample datasets so it made sense to choose the most highly represented species?
- 3) Does using *A. millepora* as the reference influence your ability to detect associations in other species? The other species are quite phylogenetically distant. I imagine using *A. millepora* biases your underlying, per-dataset association tests. And in finding shared patterns across species, 2/6 of your datasets are *A. millepora* (39% of individuals), I imagine this also slightly biases results towards *A. millepora*. I would appreciate a sentence or two in the discussion that address this but I understand that this is likely well-controlled for since you only include regions detected in 3 or more datasets (which means 2+ species).
- 4) I am not at all advocating for you to redo this work since I know how laborious it is to mine public datasets and then curate them to make them comparable, but I am curious if you thought about alternative mapping approaches here? I am wondering if more clear associations would be found if the data from each species was mapped to their respective reference (I'm not sure if *A. downingi* has a ref but the rest have high quality references) and then use something like Cactus to perform whole genome alignments and then liftover all the datasets to a single coordinate system.
- 5) Were repetitive regions masked? Spurious associations can also pop out frequently if you are getting high quality mapping to repetitive regions and I see that a number of the genes overlapping adaptive windows are associated with transposons or other repetitive proteins. I would appreciate seeing a quick mention of the how significant GEA SNPs overlaps with the repetitive regions in the assembly (if at all). The assembly is soft-masked so this should be a quick thing to check.
- 6) I would appreciate seeing something like KING-robust or other relatedness metric in addition to the $R > 0.9$ cutoff. I am not a huge fan of the $R > 0.9$ cutoff unless it's an experimentally driven threshold or if there's a good citation here. Clones and related individuals are a huge problem in any coral dataset, but could particularly affect this dataset. Related individuals are unlikely at this reefscape scale, but they would likely be $R < 0.9$ and are still good to remove. If the samples flagged by something like KING (using a relatedness threshold of say, 0.0825, the lower cutoff for second degree kin) overlap strongly with the $R > 0.9$, I would say that the $R > 0.9$ cutoff is good enough and nothing needs to be change. If they flag very different samples, I think that's a sign that the sample set may need to be revised.
- 7) I'd like to see the λ values here for the association tests of each dataset. This could be in a supplemental Q-Q plot or just the raw λ values mentioned in text. Perhaps something like this was already done, but I was unable to find it in the Zenodo. The associations for some of the datasets look very inflated, but it could just be an issue with the plotting. Even the WGS datasets show little evidence of linkage in the Manhattan plots, which makes me feel as if some of the associations within dataset could be sporadic. But again, hard for me to say from just the plot and not seeing the λ values.
- 8) Maybe I'm not understanding something from the text - Why not test all permutations of fixed effect variables? (In the "Linking environmental genomic predictions with *Acropora* cover change" section) It doesn't seem to be an unreasonably large search space to me.

In summary, Selmoni, Cleves, and Exposito-Alonso deliver an excellent manuscript that identifies adaptive windows across five *Acropora* species and uses these to project the distribution of heat resilience across global reefs under climate change. This work is novel, rigorous, and mines public datasets to draw inter-species conclusions in a way not yet done in coral literature. With minor clarifications (mostly textual and minor methodological), the work is of high merit, and I very much look forward to seeing it published. Great work, all!

(Remarks on code availability)

The code is well-documented. The README provides adequate description of data products and code used to generate the results discussed in the manuscript.

Reviewer #2

(Remarks to the Author)

This manuscript describes a study of parallel genotype-environment associations across multiple species of coral in the genus *Acropora*. Clear strengths of the study include its broad geographic scope (in terms of sampling and relevance), its use of appropriate analytical tools and procedures, and the cross-species comparison to identify genes involved in adaptation to thermal stress. It is likely that the authors have identified several genes that are conserved in function and adaptive significance among species in the genus *Acropora*, which is a notable and impactful finding. The application of that finding to models of genomic vulnerability is also likely helpful for recommending conservation actions (although the extent to which genomic information is – or should be – incorporated into actual conservation action is less clear).

I do not recommend any major revisions for this manuscript. It is largely clear, well organized, and well written. I do, nonetheless, have some recommendations for improvement that would alleviate some minor concerns about the study and the way it is presented.

First, it would be best if the authors could more clearly describe how their analysis considers the fact that there will be some overlap in GEA outliers across species by chance. Given the polygenic nature of thermal adaptation, and the inherent noise in this kind of analysis, which specific part of the between-species comparison considers the likelihood that some shared GEA outliers are shared noise (i.e. not signals of parallel adaptation). There does appear to be correction for false discovery based on the q-value method (Storey 2003), but it's unclear if this is just correcting for false-positive within-species, or if there's an additional way that random between-species overlap/concordance was corrected for. In any case, the finding that some outlier genes are found across all 7 species is likely very robust, but some clarity around these methods in the main text would be appreciated.

Second, I am wary of (maybe biased against) GO analyses. They are by nature post hoc and have the potential to be overinterpreted. The authors find that shared GEA outliers have GO terms associated with "heat shock protein binding", "unfolded protein binding", "inositol monophosphate 1-phosphatase activity", "phosphatidylinositol-4-phosphate phosphatase activity", and "glutamate receptor activity". The authors make a good case that all these GO terms are relevant to thermal adaptation, which is comforting in the context of their study. But, if I took a random selection of 85 genomic windows across the *Acropora* genome, how often would I find a gene with the phrase "heat shock" in the GO term? The authors clearly present this part of their study as exploratory – grist for future studies of the function of genes involved in heat tolerance, symbiosis, and bleaching. Also, there is no central conclusion in the manuscript that rests on the findings of this GO enrichment analysis. So, my concern is minor, but I would like to see more acknowledgement of exploratory (as opposed to conclusive) nature of this part of the study.

(Remarks on code availability)

Reviewer #3

(Remarks to the Author)

'Global coral genomic vulnerability explains recent reef losses' by Selmoni et al presents a study where they characterised genetic diversity across the *Acropora* genus and subsequently used GEA and genomic vulnerability estimates to (i) identify parallel evolutionary signals of heat adaptation (ii) predict the future distribution of heat-adapted genotypes worldwide and (iii) compare the populations that are currently protected to identify if current conservation strategies are adequate. This is an ambitious and highly interesting and innovative approach to calculating vulnerability to climate change. The authors draw on a combination of established and novel modelling techniques applied to existing genomic datasets and ecological data. The methods used were well thought out and written and conservative where appropriate. I thoroughly enjoyed the paper and recommend it for publication.

(Remarks on code availability)

Version 1:

Reviewer comments:

Reviewer #1

(Remarks to the Author)

The updated manuscript "Global coral genomic vulnerability explains recent reef losses" provides an excellent, cross-species analysis of genomic risk across pre-existing sequencing datasets and identifies genomic regions putatively responsible for recent reef declines. The authors have addressed all of my concerns from the initial review cycle and I recommend the manuscript for publication. I look forward to seeing it in print!

(Remarks on code availability)

Due to the extensive nature of the analyses, I did not download all the relevant datasets and run the code end-to-end. However, a review of the Zenodo repository suggests that the authors have well-curated their code in a reproducible manner. The repository is organized with a highly annotated master README. All bash and R scripts are named in step-by-step order and are also annotated in-line.

Reviewer #2

(Remarks to the Author)

I thank the authors for their thorough revisions in response to my (and the other reviewers') comments on an earlier version. This is a fascinating study, and I find it suitable for publication in its present form.

(Remarks on code availability)

n/a

Note: Please find attached to this resubmission an alternative version of the main text with all changes tracked in blue.

Reviewer #1

In their manuscript, Selmoni, Cleves, and Exposito-Alonso provide an excellent analysis exploring the global genomic vulnerability of reefs using publicly available genomic, photo survey, and environmental datasets. In this study, they perform GEA analyses to understand the interaction between genomic variants and the environment in six datasets covering five species of *Acropora* corals. To identify regions with conserved importance in thermal tolerance across species, they pinpoint genomic windows enriched in significant markers in three or more of the datasets. In corals, few studies have aggregated results across species in this manner to identify conserved thermotolerance markers. Using this conserved set (which includes HSP70, a gene identified in other coral stress studies), the authors build predictive models to estimate the percentage of adaptive genotypes using remotely sensed heat stress data. They then use the expected frequencies of heat-adaptive genotypes, in combination with heat stress data, to predict changes in *Acropora* percent cover, based on photo surveys from the Catlin Seaview Survey. This model is extended to project the genomic vulnerability of reef ecosystems worldwide from 2020–2040 under SSP2-4.5 climate scenarios. Reassuringly, the authors conclude that the proportion of resilient genotypes in populations will increase, although some regions are projected to experience conditions beyond the model's predictive scope (e.g., Caribbean, Persian Gulf, Red Sea). Finally, they overlay regions of predicted resilience and vulnerability onto a global map of marine protected areas (MPAs), finding that heat-resilient reefs were more frequently located within MPAs than outside of them. However, they also found that a large proportion of MPAs are protecting non-resilient reefs, a finding which highlights the need to reassess MPA design to better conserve reefs harboring heat-resilient alleles. This work is unique and makes a significant contribution to coral research. While lacking experimental data, the study is novel in aggregating results across species to predict reef-level declines. Further experimental work (which is ongoing across the world's tropical reefs) will help ground-truth this modeling framework and improve future projections. The work stands well on its own and its claims are well-supported by the literature and by the rigorous analytical framework outlined in the manuscript. As such, it provides a novel foundation for understanding how at-risk reefs may change in the future.

>>> We thank the reviewer for their careful reading of our manuscript, as well as for their positive feedback and constructive comments. We have revised the manuscript accordingly and conducted additional analyses based on the reviewer's suggestions, as detailed in the point-by-point response below.

Flaws in the manuscript are minor and primarily relate to adding language that highlights both the importance and limitations of the work. For example, the "Loss of genetic diversity in *Acropora* under global change" and "Conservation strategies for heat-adapted corals" sections could benefit from more cautionary language. While it is encouraging that allele frequencies will shift toward heat-adaptive variants, I caution that some readers may interpret this to mean that coral reefs will be fine under different climate change scenarios. Shifts in heat tolerance are likely to occur via: A) massive bottleneck events (the more likely scenario, given current emission trajectories), or B) massive reductions in emissions that allow allele frequencies to shift gradually without widespread mortality. Here would be a great place to use Florida *Acropora palmata* as case study – NOAA has an excellent report "Genotypic inventory and impact of the 2023 marine heatwave on *Acropora palmata* (elkhorn coral) populations in the Upper Florida Keys, USA: 2020-2023" by Williams et al. Pretty much all corals have died, and those that happened to survive at one restricted portion of the reef only did so because reef temperatures did not get as high.

>>> We added a paragraph to caution readers in interpreting our results. This new paragraph was placed at the end of the "Acropora heat adaptation and genomic vulnerability under future climate" section, immediately following the result that could be misinterpreted—namely that "Acropora can heat-adapt until 2040." Below is shown the new paragraph:

*"While encouraging, these results should not suggest that coral reefs will remain unaffected in the coming years. In heat-exposed *Acropora*, selection for thermally tolerant colonies will likely involve widespread mortality and recolonisation from a few remnant populations, as observed recently in Caribbean *Acropora palmata* [Williams et al., 2024]. Such turnover is likely to disrupt reef ecosystem functions, especially if other reef-building taxa respond similarly to *Acropora*. Moreover, further research is needed to determine whether local adaptation can happen under heat stress levels that exceed those observed globally today. In this regard, the only reliable solution to maintain healthy coral reefs in the long term is to limit heat stress increase by reducing carbon emission [Logan et al., 2021]."*

I really like the lines “Future adaptation to the unprecedented heat forecasted after 2040 will likely be fueled by standing genetic diversity. However, the high mortality induced by some extreme heatwaves might cause the loss of entire *Acropora* populations and the erosion of associated genetic diversity.” I’d like to see this point emphasized even stronger, particularly in the abstract and the concluding sentences of the manuscript. It’s extremely encouraging to see that heat tolerance variation exists now, but 2036 is pretty much tomorrow for these corals. Most of these species can technically reproduce at age 2 but realistically won’t until age 5-10 or more. And then from there, if reefs have declined enough, colonies may not be close enough to a viable reproductive partner for their gamete production to even be fruitful. So even if we assume an extremely liberal generation time of 2 years, this is only 5 generations. The remaining genetic diversity will be heat tolerant, but much of the reef matrix that was *Acropora* will be dead.

>>> We revised the abstract to highlight this key aspect and added the following sentences:

*“Our projections also suggest a transition where heat-adapted genotypes will spread at least until 2040. However, this transition will likely involve mass mortality of entire non-adapted populations and a consequent erosion of *Acropora* genetic diversity. This genetic diversity loss could hinder the capacity of *Acropora* to adapt to the more extreme heatwaves projected beyond 2040. Genomic vulnerability and genetic diversity loss estimates can be used to reassess which coral reefs are at risk and their conservation.”*

In addition, we modified the final sentences of the manuscript as follows:

“Our results solidify the notion that evolutionary adaptation is a key component to understanding global change-driven risks and reef losses. Because adaptation depends on standing genetic diversity, it is essential not only to protect heat-adapted genotypes but also to conserve a broad genetic diversity portfolio to support future adaptive potential. We thus advocate that reef conservation priorities should include indicators of coral genomic vulnerability and genetic diversity loss.”

I have no major methodological concerns with the analysis and believe the manuscript has strong merit for publication as-is. I also appreciate the authors making their code and datasets publicly available. The code is well annotated in the README. However, I have a few minor suggestions for clarification in the methodology:

1) I would make sure to note the kind of RAD method used for each of the datasets (rather than just the general term “RAD-Seq”), preferably in the main text, but alternatively in Table S1A. Each RAD method has its own assumptions and the research community has strong opinions on each.

>>> As suggested, we revised Table S1A to specify the RAD-seq method for each dataset.

2) In the results, it’s mentioned that *A. millepora* was used because of the quality of the reference. What defines quality? Contiguity? BUSCO? NCBI annotation? And were there other rationales as well? Was it also because you had two separate *A. millepora* sample datasets so it made sense to choose the most highly represented species?

>>> We used the *A. millepora* reference because it was the only chromosome-level, annotation-curated (RefSeq) assembly available when we performed the analysis (mid-2022 to mid-2024). New chromosome-level *Acropora* assemblies were released later throughout 2024 and 2025. We have therefore revised the sentence in the manuscript as follows:

*“[...] mapping reads from each dataset against the genome of *A. millepora* (NCBI RefSeq GCF_013753865.1; v. 2.1), which was the only chromosome-level assembly available for the genus at the time of analysis (2022-2024).”*

3) Does using *A. millepora* as the reference influence your ability to detect associations in other species? The other species are quite phylogenetically distant. I imagine using *A. millepora* biases your underlying, per-dataset association tests. And in finding shared patterns across species, 2/6 of your datasets are *A. millepora* (39% of individuals), I imagine this also slightly biases results towards *A. millepora*. I would appreciate a sentence or two in the discussion that address this but I understand that this is likely well-controlled for since you only include regions detected in 3 or more datasets (which means 2+ species).

>>> No, we did not observe more significant associations in *A. millepora* datasets. As shown in Table S1C, the number of significant associations is linked to the number of genotyped SNPs per dataset, which depends on the sequencing strategy, rather than on phylogenetic distance to the reference. We added the following sentence to emphasize this result:

“We finally used Picmin to identify 10-kb genomic windows significantly enriched ($Q < 0.1$) in heat-associated SNPs in three or more datasets (Text S5). Such overlapping signals were detected in all datasets, with their number proportional to the SNP count per dataset (Tab. S1C).”

4) I am not at all advocating for you to redo this work since I know how laborious it is to mine public datasets and then curate them to make them comparable, but I am curious if you thought about alternative mapping approaches here? I am wondering if more clear associations would be found if the data from each species was mapped to their respective reference (I'm not sure if *A. downingi* has a ref but the rest have high quality references) and then use something like Cactus to perform whole genome alignments and then liftover all the datasets to a single coordinate system.

>>> As explained above, we could not consider this option at the time of analysis because high-quality assemblies and annotations were not yet available. Even now, manually curated assemblies (RefSeq) exist only for *A. millepora* and *A. digitifera*, making the proposed approach challenging. While this approach may become feasible in the near future, it might bring limited improvements: as shown in Figure S2, mapping quality was already comparable between *A. millepora* and non-*A. millepora* species, with mapped read proportions typically exceeding 90% across all datasets.

5) Were repetitive regions masked? Spurious associations can also pop out frequently if you are getting high quality mapping to repetitive regions and I see that a number of the genes overlapping adaptive windows are associated with transposons or other repetitive proteins. I would appreciate seeing a quick mention of the how significant GEA SNPs overlaps with the repetitive regions in the assembly (if at all). The assembly is soft-masked so this should be a quick thing to check.

>>> Repetitive regions were not masked. In response to the reviewer's comment, we performed an additional analysis to test whether windows with overlapping signals were enriched in repeats. We calculated the percentage of repeats in overlapping windows and compared it to that in genomic windows genotyped in at least three datasets but lacking overlapping signals. As shown in the boxplot below, overlapping (left) and non-overlapping (right) windows exhibited similar repeat content. The code for this analysis has been added to the Zenodo repository.

6) I would appreciate seeing something like KING-robust or other relatedness metric in addition to the $R > 0.9$ cutoff. I am not a huge fan of the $R > 0.9$ cutoff unless it's an experimentally driven threshold or if there's a good citation here. Clones and related individuals are a huge problem in any coral dataset, but could particularly affect this dataset. Related individuals are unlikely at this reefscape scale, but they would likely be $R < 0.9$ and are still good to remove. If the samples flagged by something like KING (using a relatedness threshold of say, 0.0825, the lower cutoff for second degree kin) overlap strongly with the $R > 0.9$, I would say that the $R > 0.9$ cutoff is good enough and nothing needs to be change. If they flag very different samples, I think that's a sign that the sample set may need to be revised.

>>> Thank you for the suggestion of KING-robust. We run the suggested analyses. We originally used the correlation method to detect clones, as these typically underlie sampling bias where adjacent sampled corals reproduced by fragmentation. Clonal pairs are expected to have nearly identical allele frequencies, and we set the $R > 0.9$ threshold based on empirical observation. For example, the figure below illustrates allele frequency correlations among all samples in one dataset, where most pairs show low correlations, with only a single pair (presumably clones) exhibiting an anomalously high correlation (>0.9). We did not apply additional filters because the presence of genetically similar individuals on a reef can also arise from recent selective bottlenecks, which is what we wanted to characterize in downstream analyses. The distribution of these correlation values are now shown for all datasets in Fig. S9.

We also provide below the datasets filtered with our method and with KING-robust. While the latter could identify additional clones (kinship > 0.354) in a few datasets, the general patterns are the same. The results (plot below) demonstrate high concordance between the two approaches. Only in one dataset (*Acropora cervicornis*), KING-robust identified a few additional putative clones, most of which were confined to a single site where clonal replication is likely (7 clones identified). These higher kinship values may reflect localized sampling bias or could be inflated by recent genetic bottlenecks (quite likely in Caribbean *Acropora*). Given the strong agreement between the two methods and the localized nature of these additional detections, we retained the original sample set for downstream analyses. We added the code underlying this additional analysis to the Zenodo repository.

7) I'd like to see the λ values here for the association tests of each dataset. This could be in a supplemental Q-Q plot or just the raw λ values mentioned in text. Perhaps something like this was already done, but I was unable to find it in the Zenodo. The associations for some of the datasets look very inflated, but it could just be an issue with the plotting. Even the WGS datasets show little evidence of linkage in the Manhattan plots, which makes me feel as if some of the associations within dataset could be sporadic. But again, hard for me to say from just the plot and not seeing the λ values.

>>> This potential issue is already accounted for: P-values from association tests are calibrated in LFMM to control for the genomic inflation factor, and these calibrated p-values were then used in the overlap analysis to avoid potential bias. Following the reviewer's suggestion, we added information on genomic inflation factors before and after calibration to Supplementary Table S1C. For clarity, we also added a note to the "Genotype-environment association analyses" section of the Methods:
 "PicMin compared the genome-wide distribution of LFMM-adjusted P (i.e., calibrated to control for genomic inflation factor; Tab S1C) across datasets [...]"

8) Maybe I'm not understanding something from the text - Why not test all permutations of fixed effect variables? (In the "Linking environmental genomic predictions with Acropora cover change" section) It doesn't seem to be an unreasonably large search space to me.

>>> We chose not to present all possible combinations of fixed effect variables in the manuscript to maintain clarity and conciseness, particularly because interaction terms can substantially increase model complexity and reduce interpretability. Following the reviewer's suggestion, we tested all possible combinations of fixed effect variables (with and without interactions), with the results shown in the figure below. These results are consistent with those presented in the manuscript, with the best model fit (lowest AIC) achieved when including the interaction between heat stress between surveys and the expected frequencies of adaptive genotypes (DHWbtw × AGTf). The code underlying this additional analysis is available on the Zenodo repository.

In summary, Selmoni, Cleves, and Exposito-Alonso deliver an excellent manuscript that identifies adaptive windows across five Acropora species and uses these to project the distribution of heat resilience across global reefs under climate change. This work is novel, rigorous, and mines public datasets to draw inter-species conclusions in a way not yet done in coral literature. With minor clarifications (mostly textual and minor methodological), the work is of high merit, and I very much look forward to seeing it published. Great work, all!

Reviewer #1 (Remarks on code availability):

The code is well-documented. The README provides adequate description of data products and code used to generate the results discussed in the manuscript.

>>> Thank you, again, for helping improving the manuscript!

Reviewer #2

R2.1. This manuscript describes a study of parallel genotype-environment associations across multiple species of coral in the genus *Acropora*. Clear strengths of the study include its broad geographic scope (in terms of sampling and relevance), its use of appropriate analytical tools and procedures, and the cross-species comparison to identify genes involved in adaptation to thermal stress. It is likely that the authors have identified several genes that are conserved in function and adaptive significance among species in the genus *Acropora*, which is a notable and impactful finding. The application of that finding to models of genomic vulnerability is also likely helpful for recommending conservation actions (although the extent to which genomic information is – or should be – incorporated into actual conservation action is less clear).

I do not recommend any major revisions for this manuscript. It is largely clear, well organized, and well written. I do, nonetheless, have some recommendations for improvement that would alleviate some minor concerns about the study and the way it is presented.

>>> We thank the reviewer for their careful reading and constructive feedback on our work. We have revised the manuscript for clarity in response to their comments, as detailed below in our point-by-point responses.

R2.2. First, it would be best if the authors could more clearly describe how their analysis considers the fact that there will be some overlap in GEA outliers across species by chance. Given the polygenic nature of thermal adaptation, and the inherent noise in this kind of analysis, which specific part of the between-species comparison considers the likelihood that some shared GEA outliers are shared noise (i.e. not signals of parallel adaptation). There does appear to be correction for false discovery based on the q-value method (Storey 2003), but it's unclear if this is just correcting for false-positive within-species, or if there's an additional way that random between-species overlap/concordance was corrected for. In any case, the finding that some outlier genes are found across all 7 species is likely very robust, but some clarity around these methods in the main text would be appreciated.

>>> The multiple testing correction is performed on the P-values assigned to each genomic window during the between-species enrichment analysis. Following the reviewer's comment, we amended the Methods section and expanded the description of the PicMin approach to clarify how it identifies overlapping genomic regions:

"In the second step of the framework, the results of the GEA analyses were compared between datasets using PicMin¹⁷. PicMin compared the genome-wide distribution of LFMM-adjusted P (i.e., calibrated to control for genomic inflation factor; Tab S1C) across datasets. Specifically, 10-kb genomic windows were ranked in each dataset based on their LFMM-adjusted P (with top ranks corresponding to windows containing SNPs with low, i.e., significant, P). PicMin then tested for enrichment of top-ranked windows across datasets and assigned an enrichment P to each genomic window, which was then corrected for false discoveries (q-value method⁷³). Genomic windows with significant overlap were those showing (1) an enrichment $Q < 0.1$ and (2) overlap of significant GEA associations in at least three datasets."

R2.3. Second, I am wary of (maybe biased against) GO analyses. They are by nature post hoc and have the potential to be overinterpreted. The authors find that shared GEA outliers have GO terms associated with "heat shock protein binding", "unfolded protein binding", "inositol monophosphate 1-phosphatase activity", "phosphatidylinositol-4-phosphate phosphatase activity", and "glutamate receptor activity". The authors make a good case that all these GO terms are relevant to thermal adaptation, which is comforting in the context of their study. But, if I took a random selection of 85 genomic windows across the *Acropora* genome, how often would I find a gene with the phrase "heat shock" in the GO term? The authors clearly present this part of their study as exploratory – grist for future studies of the function of genes involved in heat tolerance, symbiosis, and bleaching. Also, there is no central conclusion in the manuscript that rests on the findings of this GO enrichment analysis. So, my concern is minor, but I would like to see more acknowledgement of exploratory (as opposed to conclusive) nature of this part of the study.

>>> Thank you for making this important point. GO enrichment analyses are more robust than simply counting recurring terms within an annotation set because they explicitly test whether terms are overrepresented compared to the rest of the genome. The method we used (SetRank) adds an additional layer of robustness by weighting each genomic window according to its level of significance, giving greater weight to highly significant windows in the enrichment test. This approach can therefore highlight meaningful patterns within a large number of annotations (here, 119 genes across 85 genomic windows). Nevertheless, we agree with the reviewer that GO enrichment analyses can be overinterpreted and should not be used as the basis for central conclusions. Indeed, we emphasized this point in

our recent opinion piece, *"Finding genes and pathways that underlie coral adaptation"* (Trends in Genetics, 2024).

In response to the reviewer's comment, we added a cautionary statement in this section to warn readers against over-interpreting these results:

"[...] Results from GO enrichment analyses should be interpreted cautiously and not viewed as conclusive. Nevertheless, they can be valuable for highlighting prominent candidate genes for further validation. Given their association with molecular heat responses and symbiosis, the genes underlying these functional enrichments should be prioritized in molecular and genetic studies to determine their roles in heat tolerance, symbiosis, and bleaching¹²."

Reviewer #3

R3.1 'Global coral genomic vulnerability explains recent reef losses' by Selmoni et al presents a study where they characterised genetic diversity across the Acropora genus and subsequently used GEA and genomic vulnerability estimates to (i) identify parallel evolutionary signals of heat adaptation (ii) predict the future distribution of heat-adapted genotypes worldwide and (iii) compare the populations that are currently protected to identify if current conservation strategies are adequate. This is an ambitious and highly interesting and innovative approach to calculating vulnerability to climate change. The authors draw on a combination of established and novel modelling techniques applied to existing genomic datasets and ecological data. The methods used were well thought out and written and conservative where appropriate. I thoroughly enjoyed the paper and recommend it for publication.

>>> We thank the reviewer for their thorough reading of our work and their positive feedback.